# New Derivatives of 5-Substituted Uracils: Potential Agents with a Wide Spectrum of Biological Activity

**DOI:** 10.3390/molecules27092866

**Published:** 2022-04-30

**Authors:** Vasily A. Kezin, Elena S. Matyugina, Mikhail S. Novikov, Alexander O. Chizhov, Robert Snoeck, Graciela Andrei, Sergei N. Kochetkov, Anastasia L. Khandazhinskaya

**Affiliations:** 1Engelhardt Institute of Molecular Biology, Russian Academy of Science, 119991 Moscow, Russia; vassilevs58@yandex.ru (V.A.K.); matyugina@gmail.com (E.S.M.); kochet@eimb.ru (S.N.K.); 2Department of Pharmaceutical & Toxicological Chemistry, Volgograd State Medical University, 400131 Volgograd, Russia; m-novikov1@mail.ru; 3N.D. Zelinsky Institute of Organic Chemistry, Russian Academy of Science, Leninski pr. 47, 119991 Moscow, Russia; 4Rega Institute for Medical Research, KU Leuven, B-3000 Leuven, Belgium; robert.snoeck@kuleuven.be (R.S.); graciela.andrei@rega.kuleuven.be (G.A.)

**Keywords:** 5′-norcarbocyclic nucleoside analogues, chemical synthesis, RNA viruses

## Abstract

Pyrimidine nucleoside analogues are widely used to treat infections caused by the human immunodeficiency virus (HIV) and DNA viruses from the herpes family. It has been shown that 5-substituted uracil derivatives can inhibit HIV-1, herpes family viruses, mycobacteria and other pathogens through various mechanisms. Among the 5-substituted pyrimidine nucleosides, there are not only the classical nucleoside inhibitors of the herpes family viruses, 2′-deoxy-5-iodocytidine and 5-bromovinyl-2′-deoxyuridine, but also derivatives of 1-(benzyl)-5-(phenylamino)uracil, which proved to be non-nucleoside inhibitors of HIV-1 and EBV. It made this modification of nucleoside analogues very promising in connection with the emergence of new viruses and the crisis of drug resistance when the task of creating effective antiviral agents of new types that act on other targets or exhibit activity by other mechanisms is very urgent. In this paper, we present the design, synthesis and primary screening of the biological activity of new nucleoside analogues, namely, 5′-norcarbocyclic derivatives of substituted 5-arylamino- and 5-aryloxyuracils, against RNA viruses.

## 1. Introduction

Analogues and derivatives of nucleic acid (NA) components (nucleic bases, nucleosides, and mono- and oligonucleotides) were first tested for medicinal purposes in the middle of the last century [1]. This approach is still popular today, and its possibilities are far from being exhausted. The components of NA, primarily nucleosides and nucleotides, being the main structural units of DNA and RNA, also take part in a huge number of key metabolic processes playing the role of cofactors or regulators of hundreds of reactions of various types. In this regard, even small modifications of the nucleic base or sugar fragment of the nucleoside have a significant impact on the recognition and inhibition of the respective enzymes, and thus on its activity as an anti-pathogen. Nucleic acid analogues and derivatives are currently important elements of anticancer, antiviral, and antifungal therapy [1,2,3,4,5,6].

Pyrimidine nucleoside analogues are widely used to treat infections caused by the human immunodeficiency virus (HIV) and DNA viruses from the herpes family. The mechanism of antiviral activity of nucleoside inhibitors (NIs) is intracellular phosphorylation to the corresponding analogues of nucleoside 5′-triphosphates, which then act as terminator substrates for viral polymerases. Natural substrates for polymerases of retroviruses (reverse transcriptase) and DNA viruses (DNA polymerases) are 2′-deoxyribo nucleoside-5′-triphosphates. And only certain modifications of the pyrimidine base and the 2′-deoxyribose fragment make it possible to preserve the possibility of intracellular phosphorylation and manifestation of activity by this mechanism. Other pyrimidine derivatives belong to the class of non-nucleoside inhibitors (NNIs) and are capable of inhibiting viral polymerases by a noncompetitive mechanism [7,8,9]. These compounds bind to polymerases outside the active site, are hydrophobic, and are structurally much less similar to the natural components of NA. 

Among the 5-substituted pyrimidine nucleosides, there are not only the classical NIs of the herpes family viruses, 2′-deoxy-5-iodocytidine [10,11] and brivudine (BVDU, 5-bromovinyl-2′-deoxyuridine) [12,13], but also derivatives of 1-(benzyl)-5-(phenylamino)uracil, which proved to be non-nucleoside inhibitors of HIV-1 and EBV [14] (Figure 1). Rigid amphipathic fusion inhibitors (RAFIs) have also been described [15,16,17], which inhibit unrelated enveloped viruses such as influenza, HCV, VSV, HSV-1, HSV-2, mCMV, and TBEV without cytotoxic or cytostatic effects (SI > 3000). The mechanism of action of RAFIs is biophysical, as they interact with virion envelope lipids and prevent the fusion of the viral and cell membranes by inhibiting the increased negative curvature required for the initial stages of fusion.

The antibacterial activity of 5-substituted pyrimidine analogues of nucleosides was revealed later [6]. Analogues with extended 1-alkynyl substituents demonstrated inhibitory activity against *M. tuberculosis*, *M. avium*, and *M. bovis* in vitro [5,18]. The study of the effect of the modification of 5-(1-alkynyl)pyrimidine nucleoside analogues carbohydrate fragment on anti-mycobacterial properties showed that changes in the carbohydrate fragment, including its replacement with acyclic or carbocyclic residues, do not affect the ability to inhibit the growth of mycobacteria [19,20,21,22,23]. The process of growth inhibition of *M. tuberculosis* (H37Rv strain) was accompanied by the accumulation of lipid intracellular vacuole-like inclusions in cells, the appearance of deep protrusions and depressions on the surface, and partial and/or complete destruction of the three-layer cell membrane [24]. The ability to inhibit the growth of laboratory and MDR strains of *M. tuberculosis* has also been described for 5′-norcarbocyclic derivatives of substituted 5-arylaminouracils [25].

Thus, it has been shown that 5-substituted uracil derivatives can inhibit HIV-1 herpes family viruses, mycobacteria and other pathogens through various mechanisms. It made this modification of nucleoside analogues very promising in connection with the emergence of new viruses and the crisis of drug resistance, when the task of creating effective antiviral agents of new types that act on other targets or exhibit activity by other mechanisms is very urgent. In this paper, we present the synthesis and primary screening of the biological activity of new nucleoside analogues, namely, 5′-norcarbocyclic derivatives of substituted 5-arylamino- and 5-aryloxyuracils, against RNA viruses.

## 2. Results and Discussion

Carbocyclic nucleosides are analogues of natural nucleosides in which the oxygen atom of the furanose ring is replaced by a methylene group. These compounds are highly stable against the pseudoglycoside bond cleavage reaction induced by phosphorylases and hydrolases compared to ribonucleoside analogues and exhibit a wide spectrum of biological activity [26,27,28,29,30,31]. A feature of the structure of 5′-norcarbocyclic analogues is the absence of a 5′-methylene group. The replacement of the hydroxymethyl (-CH_2_OH) group containing the primary hydroxyl with a secondary hydroxyl deprives the 5′-norcarbocyclic nucleoside analogues of the ability to be converted to phosphorylated forms by cellular enzymes. This feature is useful in studying the mechanism of action of compounds, since it allows one to determine the need for intracellular phosphorylation to realize the biological effect of the drug and to exclude some targets traditional for conventional modified nucleosides.

### 2.1. Synthesis of Target Compounds

Figure 1 shows the preparation of a series of new 5-amino derivatives of uracil and their 5′-norcarbocyclic analogues. This synthesis of modified pyrimidines is based on the well-known bromine atom substitution reaction in 5-bromouracil. Thus, the literature describes the condensation of 5-bromouracil with amines of high nucleophilicity: aliphatic (n-butylamine, morpholine), alicyclic (cyclohexylamine) and aromatic fatty (benzylamine, *N*-methylbenzylamine). The reaction was carried out in the medium of an aminating agent at reflux for 1–2 h. The yields of target 5-amino derivatives of uracil were in the range of 75–100% [32]. The reaction time was reduced by an order of magnitude by using microwave irradiation of the reaction medium. However, in this case, the reaction was also carried out in the medium of an amination agent [33]. To obtain 5-phenylamino derivatives of uracil, Gerns and Perrota treated 5-bromouracil with a three-fold molar excess of the corresponding aniline in refluxing ethylene glycol, which was used as a solvent [34]. The use of ethylene glycol as a solvent was also successfully used in the amination of 1-benzyl-5-bromo derivatives of uracil with a 3.6-fold molar excess of a fatty aromatic amine (benzylamine and phenethylamine). The yield of target 5-benzylamino- and 5-phenethylamino- derivatives of uracil was in the range of 70–75% [14].

In order to reduce the consumption of amine, a 1.5-fold molar excess of quinoline as an acceptor of hydrogen bromide released during the reaction was added into the reaction mixture containing 2-fold molar excess of *p*-substituted aniline **1d–i**, 5-bromouracil and ethylene glycol as a solvent. As a result, 5-arylaminouracils **2d–i** were obtained in 68–80% yields. The corresponding 5-(azepan-1-yl)uracil **2a**, 5-(4-phenylpiperazin-1-yl)uracil **2b**, as well as 5-(3,4-dihydroisoquinolin-2(1H)-yl)uracil **2c** were prepared under the same conditions using azepane **1a**, 1-phenylpiperazine **1b**, and 1,2,3,4-tetrahydroisoquinoline **1c**. For NMR spectra of compounds **2a–i** see Appendix A in Appendix A.

5-Aminouracils **2a–i** were then subjected to condensation with 6-oxobicyclo[3.1.0]hex-2-ene (1.3 molar equivalent) in the presence of tetrakis(triphenylphosphine)palladium(0) as a catalyst according to the Trost procedure [35] to give mono- and disubstituted 5′-norcarbocyclic derivatives, (±)**-3a–i** (in yields of 14–35%) and (±)-**4a–i** (in yields of 21–41%), respectively. The products **3a–i** and **4a–i** (see Appendix A Appendix A) were obtained as racemic mixtures.

It is important to note that the use of a 1.3 molar equivalent of 6-oxobicyclo[3.1.0]hex-2-ene turned out to be the optimal solution for obtaining both target substances at the same time. A further increase in the molar ratio of 6-oxobicyclo[3.1.0]hex-2-ene and the modified pyrimidine base, the main product of the reaction becomes 1,3-di-(4-hydroxycyclopent-2-en-1-yl)-5-substituted uracil, and the 1-(4-hydroxycyclopent-2-en-1-yl)-5-substituted product is observed in trace amounts.

Figure 2 illustrates the synthesis of 5-aryloxyuracils **5a–d** by the methods described previously [36,37,38]. The starting 4-substituted phenols **6a–d** were treated with ethyl bromoacetate in a DMF solution in the presence of K_2_CO_3_ to give the corresponding phenoxyacetic acid ethyl esters **7a–d**. The esters **7a–d** were treated with ethyl formate in a solution of THF or diethyl ether in the presence of NaH. The resulting formyl derivatives **8a–8d** without additional purification were subjected to condensation with a 1.5-fold molar excess of thiourea in an anhydrous isopropanol solution in the presence of NaH to give the corresponding 5-phenoxy-2-thiouracils **9a–d**. To obtain 5-phenoxyuracils **5a–d**, 2-thiouracils **9a–d** were subjected to desulfurization in an aqueous medium in the presence of a four-fold molar excess of monochloroacetic acid (MCA) and hydrochloric acid. The yields of 5-phenoxyuracils **5a–d** (see Appendix A in Appendix A) were of 77–85%.

It should be noted that the use of potassium *tert*-butoxide (*t*BuOK) instead of NaH reduces the yield of compounds **5a–d** to 17%. The stage of synthesis of mono-**10a–d** and disubstituted **11a–d** 5′-norcarbocyclic derivatives of 5-phenoxyuracils **5a–d** (Figure 2) is completely identical to the procedure described above for the preparation of 5′-norcarbocyclic derivatives of 5-aminouracils **3a–i** and **4a–i** (Figure 1). The yields of the racemic target products (±)-**10a–d** and (±)-**11a–d** were 18–53% and 23–34%, respectively (see Appendix A in Appendix A).

High regio and stereoselectivity of the Trost reaction is achieved due to the unique structure of (π-allyl)palladium complex that the 6-oxobicyclo[3.1.0]hex-2-ene forms with tetrakis(triphenylphosphine)palladium(0) [39]. The polar hydroxy group provides the regiocontrolled nucleophilic addition to the π-allyl system. The structure of the resulting palladium complex is such that the nucleophile adds to the carbocyclic moiety from the side opposite to palladium, to give only cis-products, thus providing the stereospecificity of the reaction. The formation of a monosubstituted product at the N-1 position is also due to the peculiarities of the Trost reaction mechanism. Attack of the complex with N-1 nitrogen is preferable, since the nucleophilicity of N-3 uracil is negatively affected by the presence of two neighboring carbonyl groups with pronounced electron-withdrawing properties. This pattern is generally observed for all types of N-substitution involving uracil and its derivatives [40,41]. On the other hand, the formation of N-1,3 substitution products occurs, apparently, due to the attack of the resulting N-1 nucleoside on the second epoxide molecule.

The use of 6-oxobicyclo[3.1.0]hex-2-ene in the Trost condensation affords racemic products, which can be used for primary activity screening. Individual enantiomers of the most active compounds can be latter catalytically cleaved using various lipases [42,43].

### 2.2. Biological Activity

Broad-spectrum antiviral activity screening was performed. Antiviral activity and cytotoxicity were assessed for all new compounds in cell-based assays against a variety of RNA viruses. The data in Table 1 show the presence of an antiviral effect against the causative agent of SARS-HCoV OC43. Mono- and disubstituted 5′-norcarbocyclic derivatives of 5-arylaminouracils (±)-**3f**, (±)-**3i** and (±)-**4d**, with an extended alkoxy or alkyl group in the *p*-position of the aryl fragment, and disubstituted aryloxyuracil derivative (±)-**11a** have moderate biological activity against this pathogen. Nucleoside analogue (±)-**11a** was the only compound that inhibited the respiratory syncytial virus RSV (strain A Long). 1,3-Bis-5′-norcarbocyclic derivative of 5-aryloxyuracial (±)-**11d** demonstrated activity against the yellow fever virus YFV (strain 17D). Testing of the other representatives of the coronavirus family (HCoV-229E, NL63), a number of influenza A virus serotypes (Influenza H1N1, H3N2) and influenza B virus showed no activity against these pathogens. None of the 5-aminouracils **2a–i** and 5-phenoxyuracils **5a–d** demonstrated either any antiviral activity or cytotoxicity.

It should be noted that although most of the compounds that showed antiviral activity on a particular infected cell line had no toxicity against that cell line up to concentration of 100 μM, a marked cytotoxicity was found in at least one of the other tested cell lines. Compounds (±)-**3d**, (±)-**3e**, (±)-**4e**, (±)-**10a** and (±)-**11c** did not show any activity but had marked cytotoxicity in at least one of the three tested cell lines.

Some of the synthesized compounds were additionally tested in HEL cells infected with the DNA viruses: herpes viruses (HSV-1 KOC), varicella-zoster virus (strains TK- and TK+) and cytomegalovirus (strains AD169 and Davis). Among them, only compound (±)-**4e** showed inhibitory activity. It inhibited the formation of viral plaques in the TK-varicella-zoster virus strain (EC_50_ = 48.89 μM) and in both cytomegalovirus strains (EC_50_ = 76.47 and 40.90 μM, respectively). The minimum concentration of compound (±)-**4e** that leads to the detection of changes in cell morphology was greater than 100 μM.

Thus, it can be assumed that the 5′-norcarbocyclic fragment of 5-substituted uracil derivatives plays an important role and is required for the biological activity. It has been shown that 5′-norcarbocyclic analogues of 2′,3′-dideoxy-2′,3′-didehydrouridine with different substituents in the 5 position of the heterocyclic base have activity against various viruses. To conclude, 5′-norcarbocyclic derivatives of 5-substituted uracils have a wide spectrum of activity and are promising for a more detailed study of the structure activity relationship.

## 3. Materials and Methods

### 3.1. Chemistry

All reagents used were of high quality, are commercially available, and do not require further purification (unless otherwise noted). Ethylene glycol, isopropanol, and DMF were supplied by Sigma-Aldrich Co. (Saint Louis, MO, USA). Anhydrous 1,2-dichloroethane and ethyl acetate was obtained by distillation over P_2_O_5_. The tetrakis(triphenylphosphine)palladium(0) catalyst was purchased from Alfa Aesar (Ward Hill, MA, USA). 6-Oxobicyclo[3.1.0]hex-2-ene was synthesized based on a previously published procedure [44].

Column chromatography was performed on Silica Gel 60 0.040–0.063 mm (Merck, Germany). Thin layer chromatography (TLC) was carried out on TLC Silica gel 60 F_254_ plates (Merck, Germany). For preparative thin-layer chromatography (PLC), PLC Silica gel 60 F_254_, 1 mm plates (Merck, Germany) were used. The results of separation on the plate, as well as the determination of the retention factor (Rf), were analyzed using a VL-6.LC UV lamp (Vilber Lourmat Deutschland, Eberhardzell, Germany). The Rf parameter was measured in the solvent system CHCl_3_: MeOH (95:5). Eluent systems for the separation of substances are indicated in the text.

NMR spectra were recorded on a Bruker Avance 400 spectrometer (400 MHz for ^1^H NMR, 100 MHz for ^13^C NMR) in CDCl_3_, CD_3_OD, a mixture of CD_3_OD/CDCl_3_, or DMSO-d6 containing tetramethylsilane as an internal standard. Chemical shifts are given in ppm. The product yields refer to spectroscopically (^1^H NMR and ^13^C NMR) homogeneous substances.

Melting points were determined using a Mel-Temp 3.0 (Laboratory Devices Inc., Auburn, CA, USA).

High resolution mass spectra were measured on a Bruker Daltonics micrOTOF II or maXis (Bruker, Germany) instruments using electrospray ionization (ESI HRMS). The measurements were done in a positive ion mode with the corresponding parameters: interface capillary voltage 4500 V, mass range from *m*/*z* 50 to 3000; internal calibration was done with ESI Tuning Mix (Agilent, Merck, Germany). A syringe injection was used for solutions in acetonitrile (flow rate 3 mL min^−1^). Nitrogen was applied as a drying gas (4.0 L/min); the interface temperature was set at 180 °C.

#### 3.1.1. General Procedure for the Preparation of 5-Aminouracil Derivatives (**2a–i**)

A mixture of 5-bromouracil (2.0 g, 10.47 mmol), the corresponding amine **1a–i** (20.95 mmol), quinoline (1.9 mL, 16.08 mmol) and of ethylene glycol (50 mL) was refluxed for 1 h, cooled to room temperature, and cold water (250 mL) was added to the solidified reaction mass. The precipitate was filtered off, washed on the filter with water (3 × 50 mL) to remove ethylene glycol residues and ethyl acetate (3 × 15 mL) to remove traces of quinoline, unreacted amine, and its oxidation products, and dried on a Petri dish at 70°C for several hours. The raw material obtained was recrystallized from aqueous DMF.

5-(Azepan-1-yl)uracil **2a** (1.79 g, 8,48 mmol, 81%). Decomposition at 310 °C. Rf 0.33. ^1^H NMR (DMSO-d6) δ, ppm: 10.92 (1H, s, HN^3^), 10.27 (1H, s, HN^1^), 6.63 (1H, s, H-6), 2.99–2.95 (4H, m, 2′-CH_2_, 7′-CH_2_), 169–1.64 (4H, m, 3′-CH_2_, 6′-CH_2_), 1.56–1.52 (4H, m, 4′-CH_2_, 5′-CH_2_). ^13^C NMR (DMSO-d6) δ, ppm: 162.4, 150.6, 128.0, 124.4, 52.0 × 2, 29.1 × 2, 27.3 × 2.

5-(4-Phenylpiperazin-1-yl)uracil **2b** (2.25 g, 8.27 mmol, 79%). M.p. 352–354.5 °C. Rf 0.28. ^1^H NMR (DMSO-d6) δ, ppm: 10.72 (1H, s, HN^3^), 10.17 (1H, s, HN^1^), 7.25–7.20 (2H, m, H-3″, H-5″), 6.98–6.94 (2H, m, H-6, H-4″), 6.82–6.77 (2H, m, H-2″, H-6″), 3.25–3.21 (4H, m, 2′-CH_2_, 6′-CH_2_), 3.04–2.99 (4H, m, 3′-CH_2_, 5′-CH_2_). ^13^C NMR (DMSO-d6) δ, ppm: 161.9, 151.6, 150.7, 129.4 × 2, 127.0, 126.6, 119.41, 116.1 × 2, 50.2 × 2, 49.0 × 2.

5-(3,4-Dihydroisoquinolin-2(1H)-yl)uracil **2c** (1.91 g, 7.85 mmol, 75%), M.p. 311–312 °C. Rf 0.29. ^1^H NMR (DMSO-d6) δ, ppm: 10.64 (1H, s, HN^3^), 10.11 (1H, s, HN^1^), 7.16–7.06 (4H, m, H-5′, H-6′, H-7′, H-8′), 6.81 (1H, s, H-6), 4.06 (2H, s, 2H-1′), 3.24–3.20 (2H, s, 2H-3′), 2.89–2.85 (2H, s, 2H-4′). ^13^C NMR (DMSO-d6) δ, ppm: 162.1, 150.8, 134.8, 134.2, 129.1, 127.1, 126.9, 126.7, 126.6, 126.1, 52.6, 47.9, 28.9.

5-((4′-Hexylphenyl)amino)uracil **2d** (2.11 g, 7.33 mmol, 70%). M.p. 332–333 °C, Rf 0.38. ^1^H NMR (DMSO-d6) δ, ppm: 10.85 (1H, s, HN^3^), 10.14 (1H, s, HN^1^), 7.14 (1H, s, H-6), 6.99–6.96 (2H, m, H-3′, H-5′), 6.77–6.74 (2H, m, H-2′, H-6′), 2.52–2.46, 1.60–1.50 (2H, m, α-CH_2_), (2H, m, β-CH_2_), 1.34–1.30 (6H, m, (CH_2_)_3_), 0.90–0.86 (3H, m, CH_3_). ^13^C NMR (DMSO-d6) δ, ppm: 162.6, 150.5, 143.8, 133.3, 129.0 × 2, 129.0, 117.7, 116.0 × 2, 34.8, 31.5, 31.3, 28.7, 22.3, 14.1.

5-((4′-tert-Butylphenyl)amino)uracil **2e** (1.85 g, 7.12 mmol, 68%). M.p. 354.5–356.5 °C. Rf 0.27. ^1^H NMR (DMSO-d6) δ, ppm: 10.93 (1H, s, HN^3^), 10.22 (1H, s, HN^1^), 7.20–7.15 (3H, m, H-3′, H-5′, H-6), 6.79–6.74 (2H, m, H-2′, H-6′), 1.26 (9H, s, 3CH_3_). ^13^C NMR (DMSO-d6) δ, ppm: 162.7, 150.6, 143.6, 141.7, 129.5, 125.9 × 2, 117.5, 115.5 × 2, 34.1, 31.8 × 3.

5-((4′-Heptylphenyl)amino)uracil **2f** (2.24 g, 7.43 mmol, 71%). M.p. 328–330 °C. Rf 0.28. ^1^H NMR (DMSO-d6) δ, ppm: 10.92 (1H, s, HN^3^), 10.21 (1H, s, HN^1^), 7.14 (1H, s, H-6), 6.99–6.95 (2H, m, H-3′, H-5′), 6.77–6.72 (2H, m, H-2′, H-6′), 6.42 (1H, s, HN^5^), 2.50–2.45 (2H, m, α-CH_2_), 1.59–1.49 (2H, m, β-CH_2_), 1.33–1.24 (8H, m, (CH_2_)_4_), 0.90–0.86 (3H, m, CH_3_). ^13^C NMR (DMSO-d6) δ, ppm: 162.6, 150.6, 143.8, 133.1, 129.3, 129.1 × 2, 117.6, 115.9 × 2, 34.8, 31.6, 31.5, 29.0, 28.9, 22.4, 14.2.

5-((4′-iso-Propylphenyl)amino)uracil **2g** (1.95 g, 7.96 mmol, 76%). M.p. 334.5–345 °C. Rf 0.29. ^1^H NMR (DMSO-d6) δ, ppm: 10.93 (1H, s, HN^3^), 10.22 (1H, s, HN^1^), 7.16 (1H, s, H-6), 7.05–7.00 (2H, m, H-3′, H-5′), 6.78–6.73 (2H, m, H-2′, H-6′), 6.42 (1H, s, HN^5^), 2.84–2.75 (1H, m, *t*CH), 1.18 (6H, d, (CH_3_)_2_). ^13^C NMR (DMSO-d6) δ, ppm: 162.7, 151.0, 143.9, 139.3, 129.4, 127.0 × 2, 117.6, 115.9 × 2, 33.0, 24.4 × 2.

5-((4′-Hexyloxyphenyl)amino)uracil **2h** (2.22 g, 7.33 mmol, 70%) M.p. 322–324 °C. Rf 0.34. ^1^H NMR (DMSO-d6) δ, ppm: 10.92 (1H, s, HN^3^), 10.13 (1H, s, HN^1^), 7.02 (1H, s, H-6), 6.83–6.75 (4H, m, H-3′, H-5′, H-2′, H-6′), 6.25 (1H, s, HN^5^), 3.92–3.87 (2H, m, α-CH_2_), 1.73–1.64 (2H, m, β-CH_2_), 1.48–1.40 (2H, m, γ-CH_2_), 1.38–1.30 (4H, m, (CH_2_)_2_), 0.92–0.87 (CH_3_). ^13^C NMR (DMSO-d6) δ, ppm: 162.5, 153.1, 150.4, 139.1, 126.5, 118.8, 118.2, 116.1 × 2, 68.8, 31.4, 29.3, 25.6, 22.4, 14.1.

5-((4′-Heptyloxyphenyl)amino)uracil **2i** (2.66 g, 8.38 mmol, 80%). Rf 0.34. ^1^H NMR (DMSO-d6) δ, ppm: 10.92 (1H, s, HN^3^), 10.13 (1H, s, HN^1^), 7.02 (1H, s, H-6), 6.84–6.75 (4H, m, H-3′, H-5′, H-2′, H-6′), 6.26 (1H, s, HN^5^), 3.92–3.87 (2H, m, α-CH_2_), 1.73–1.64 (2H, m, β-CH_2_), 1.47–1.28 (8H, m, γ-CH_2_, (CH_2_)_3_), 0.91–0.87 (CH_3_). ^13^C NMR (DMSO-d6) δ, ppm: 162.5, 153.1, 150.4, 139.1, 126.5, 118.8, 118.2 × 2, 116.1 × 2, 68.8, 31.6, 29.4, 28.8, 25.9, 22.3, 14.1.

#### 3.1.2. General Procedure for the Preparation of 5-(Phenoxy)-2-Thiouracils **9a–d**

Potassium carbonate (13.8 g, 99.85 mmol) and ethyl bromoacetate (8.5 mL, 76.65 mmol) were added to a solution of phenol 6a–d (69.40 mmol) in DMF (50 mL). The resulting mixture was stirred at 80 °C for 24 h, evaporated to dryness in vacuum, and the residue was dissolved in chloroform (100 mL) and washed with a 5% NaOH solution (3 × 50 mL) and water (100 mL). The organic fraction was dried over anhydrous Na_2_SO_4_, filtered and evaporated under reduced pressure. The resulting ethers 7a–d (45.02 mmol) was dissolved in THF (50 mL) without additional purification, ethyl formate (111 mmol) and NaH (113 mmol, a 60% suspension in oil) were added. The resulting mixture was stirred at room temperature overnight, the solvent was evaporated under reduced pressure, the residue was dissolved in anhydrous *i*PrOH (80 mL) and then thiourea (67.00 mmol) and NaH (66.67 mmol, a 60% suspension in oil) were added. About 30 min latter, after the end of gas evolution, the reaction mixture was refluxed for 24 h. Then it was evaporated to dryness, dissolved in 250 mL of water, acidified with hydrochloric acid to pH 2, and the precipitate formed was filtered off, dried in air and recrystallized from a mixture of DMF and water (1:1).

5-(3,5-Dimethylphenoxy)-2-thiouracil **9a**. Yield 83%. M.p. 242–244 °C, Rf 0.60 (dioxane-NH_4_OH, 4:1). ^1^H NMR (DMSO-d6) δ, ppm: 12.71 (1H, s, HN^3^), 12.33 (1H, s, HN^1^), 7.52 (1H, s, H-6), 6.69–6.52 (4H, s, H-3′, H-5′, H-2′, H-6′), 2.23 (6H, s, 2CH_3_). ^13^C NMR (DMSO-d6) δ, ppm: 174.6, 157.9, 157.7, 139.4 x2, 134.5, 133.6, 124.6, 113.6 × 2, 21.3 × 2.

5-(4-Butoxyphenoxy)-2-thiouracil **9b**. Yield 88%. M.p. 233–236 °C, Rf 0.76 (dioxane-NH_4_OH, 4:1). ^1^H NMR (DMSO-d6) δ, ppm: 12.71 (1H, s, HN^3^), 12.28 (1H, s, HN^1^), 7.43 (1H, s, H-6), 6.96–6.84 (4H, m, H-3′, H-5′, H-2′, H-6′), 3.93–3.89 (2H, m, α-CH_2_), 1.72–1.62 (2H, m, β-CH_2_), 1.49–1.36 (2H, m, γ-CH_2_), 0.95–0.89 (3H, m, CH_3_). ^13^C NMR (DMSO-d6) δ, ppm: 174.3, 157.9, 154.9, 151.1, 135.7, 132.4, 117.6 × 2, 115.7 × 2, 68.1, 31.3, 19.2, 14.2.

5-(4-Chlorophenoxy)-2-thiouracil **9c**. Yield 76%. M.p. 206–209 °C, R**f** 0.62 (dioxane-NH_4_OH, 4:1). ^1^H NMR (DMSO-d6) δ, ppm: 13.02 (1H, s, HN^3^), 7.36–7.30 (2H, m, H-3′, H-5′), 6.97–6.92 (2H, m, H-2′, H-6′), 4.69 (1H, s, H-6). ^13^C NMR (DMSO-d6) δ, ppm: 170.4, 157.1, 129.7 × 2, 129.2, 116.8 × 2, 65.2.

#### 3.1.3. General Procedure for the Preparation of 5-(Phenoxy)uracils **5a–d**

2-Thiouracil **9a–d** (10.07 mmol) was mixed with a solution of MCA (3 g, 31.75 mmol) in a mixture of water (30 mL) and concentrated hydrochloric acid (10 mL), the resulting mixture was refluxed for 36 h, filtered, the solid residue was repeatedly washed with water, dried in air and recrystallized from a mixture of DMF and water (2:1).

5-(3,5-Dimethylphenoxy)uracil **5a** (13.5 g, 58.3 mmol, 84%). M.p. 293–295 °C, Rf 0.27. ^1^H NMR (DMSO-d6) δ, ppm: 11.34 (1H, d, HN^3^), 10.79 (1H, m, HN^1^), 7.55 (1H, s, H-6), 6.64 (1H, s, H-4), 6.56 (2H, s, H-2′, H-6′), 2.22 (6H, s, 2CH_3_). ^13^C NMR (DMSO-d6) δ, ppm: 160.6, 158.5, 151.2, 139.2 × 2, 133.9, 129.4, 124.1, 113.2 × 2, 21.4 × 2.

5-(4-Butoxyphenoxy)uracil **5b** (15.3 g, 55.5 mmol, 80%), M.p. 271–274 °C. Rf 0.27. ^1^H NMR (DMSO-d6) δ, ppm: 11.34–11.33 (1H, d, HN^3^), 10.77–10.75 (1H, m, HN^1^), 7.56–7.51 (1H, d, H-6), 6.96–6.82 (4H, m, H-3′, H-5′, H-2′, H-6′), 3.94–3.88 (2H, m, α-CH_2_), 1.71–1.62 (2H, m, β-CH_2_), 1.49–1.36 (2H, m, γ-CH_2_), 0.95–0.90 (3H, m, CH_3_). ^13^C NMR (DMSO-d6) δ, ppm: 160.7, 154.4, 152.2, 151.1, 133.5, 130.3, 116.7 × 2, 115.6 × 2, 68.0, 31.3, 19.2, 14.2.

5-(4-Chlorophenoxy)uracil **5c** (14.1 g, 59.0 mmol, 85%). M.p. 331–333 °C, Rf 0.24. ^1^H NMR (DMSO-d6) δ, ppm: 11.40 (1H, d, HN^3^), 10.88–10.85 (1H, m, HN^1^), 7.68–7.66 (1H, d, H-6), 7.36–7.30 (2H, m, H-3′, H-5′), 7.03–6.98 (2H, m, H-2′, H-6′). ^13^C NMR (DMSO-d6) δ, ppm: 160.4, 157.3, 151.1, 134.4, 129.7 × 2, 129.2, 126.3, 117.4 × 2.

5-(4-tert-Butylphenoxy)uracil **5d** (12.6 g, 48.9 mmol, 70%), M.p. 273–275 °C, Rf 0.27. ^1^H NMR (DMSO-d6) δ, ppm: 11.37–11.36 (1H, d, HN^3^), 10.82–10.79 (1H, m, HN^1^), 7.59–7.57 (1H, d, H-6), 7.32–7.27 (2H, m, H-3′, H-5′), 6.89–6.83 (2H, m, H-2′, H-6′), 1.26 (9H, s, (CH_3_)_3_). ^13^C NMR (DMSO-d6) δ, ppm: 160.7, 156.3, 151.1, 144.8, 134.0, 129.5, 126.6 × 2, 115.0 × 2, 34.3, 31.8 × 3.

#### 3.1.4. General Procedure for the Preparation of Mono- and Di-Substituted 5′-Norcarbocyclic Derivatives of 5-Aminouracils ((±)-**3a–i**, (±)-**4a–i**) and 5-Phenoxyuracils ((±)-**10a–d**, (±)-**11a–d**)

The corresponding 5-substituted uracil derivative (100 mg) was dissolved in dry DMF (10 mL) and dried under vacuum to remove possible traces of water. This procedure was repeated twice. 6-Oxobicyclo[3.1.0]hex-2-ene (1.3 molar equivalents) was dissolved in freshly distilled THF (3 mL) and added to the resulting suspension in DMF, the reaction mixture was purged with argon and Pd(PPh_3_)_4_ (0.07 molar equivalents) was finally added. The reaction mixture was stirred at room temperature overnight, and then evaporated to dryness under vacuum. The target compounds were isolated by column chromatography on silica gel. The solid products were recrystallized from a mixture of CHCl_3_ and ethyl acetate.

(±)-1-(4′-Hydroxycyclopent-2′-en-1′-yl)-5-(azepan-1″-yl)uracil **3a** and (±)-1-(4′-hydroxycyclopent-2′-en-1′-yl)-3-(4‴-hydroxycyclopent-2‴-en-1‴-yl)-5-(azepan-1″-yl)uracil 4a were synthesized from starting 5-(azepan-1-yl)uracil **2a**. Compound **3a** was isolated by silica gel column chromatography in CHCl_3_:MeOH (98:2) eluent system. Remaining impurities were removed by PLC in CHCl_3_:MeOH (95:5) eluent system. The target 3a was obtained as yellow oil (47 mg, 0.16 mmol, 33.3%). Rf 0.30. ^1^H NMR (CDCl_3_:CD_3_OD, 3:1) δ, ppm: 6.91 (1H, m, H-6), 6.19–6.17 (1H, m, H-2′), 5.80–5.78 (1H, m, H-3′), 5.52–5.49 (1H, m, H-1′), 4.78–4.75 (1H, m, H-4′), 3.10 (4H, s, 2H-2″, 2H-7″), 2.86–2.76 (1H, m, αH-5′), 1.76 (4H, s, 2H-3″, 2H-6″), 1.55–1.52 (1H, m, βH-5′). ^13^C NMR (CDCl_3_) δ, ppm: 161.2, 149.9, 139.2, 132.0, 129.4, 122.9, 74.6, 59.7, 52.0 × 2, 39.6, 29.7×2, 28.8 × 2, HRMS, *m*/*z*: calculated for C_15_H_21_N_3_O_3_ [M + H]^+^ 292.1656; found 292.1655.

Compound **4a** was isolated by column chromatography on silica gel in a CHCl_3_:MeOH (99:1) eluent system and then purified by PLC in a CHCl_3_:MeOH (95:5) system. The resulting compound was obtained as yellow oil (47.5 mg, 0.13 mmol, 27%). Rf 0.48. ^1^H NMR (CDCl_3_) δ, ppm: 6.22–6.20 (1H, m, H-2⁗), 6.12–6.09 (1H, m, H-2′), 5.98–5.94 (1H, m, H-3⁗), 5.82–5.80 (1H, m, H-3′), 5.77–5.74 (1H, m, H-1⁗), 5.60–5.57 (1H, m, H-1′), 4.87–4.83 (1H, m, H-4⁗), 4.73–4.67 (1H, m, H-4′), 3.07–3.04 (4H, m, 2H-2″, 2H-7″), 2.88–2.70 (2H, m, αH-5′, αH-5⁗), 1.98–1.91 (1H, m, βH-5⁗), 1.80–1.75 (4H, s, 2H-3″, 2H-6″), 1.64–1.56 (5H, m, 2H-4″, 2H-5′, βH-5′). ^13^C NMR (CDCl_3_) δ, ppm: 161.0, 149.9, 139.3, 136.7, 132.1, 131.2, 128.8, 122.8, 76.2, 74.6, 60.0, 56.3, 52.3 × 2, 39.9, 37.5, 29.7 × 2, 28.4×2. HRMS, *m*/*z*: calculated for C_24_H_28_N_4_O_4_ [M + H]^+^ 437.2183; found 437.2180.

(±)-1-(4′-Hydroxycyclopent-2′-en-1′-yl)-5-(4″-phenylpiperazin-1″-yl)uracil **3b** and (±)-1-(4′-hydroxycyclopent-2′-en-1′-yl)-3-(4‴-hydroxycyclopent-2‴-en-1‴-yl)-5-(4″-phenylpiperazin-1″-yl)uracil **4b** were synthesized from the starting 5-(4-phenylpiperazin-1-yl)uracil **2b**. Compound **3b** was isolated by column chromatography on silica gel in CHCl_3_:MeOH eluent system (97:3), and then purified by PLC in CHCl_3_:7M NH_3_/MeOH (9:1). The target **3b** was obtained as a gray powder (21.2 mg, 0.06 mmol, 16.2%). Decomposed at 235 °C. Rf 0.35. ^1^H NMR (CDCl_3_:CD_3_OD, 3:1) δ, ppm: 7.32–7.27 (2H, t, H-3‴, H-5‴), 7.09–7.03 (3H, s, H-6, H-2‴, H-6‴), 6.98–6.93 (1H, t, H-4‴), 6.23–6.20 (1H, m, H-2′), 5.83–5.80 (1H, m, H-3′), 5.58–5.54 (1H, m, H-1′), 4.80–4.76 (1H, m, H-4′), 3.35–3.32 (4H, m, 2H-3″, 2H-5″), 3.14–3.11 (4H, m, 2H-2″, 2H-6″), 2.86–2.76 (1H, m, αH-5′), 1.63–1.55 (1H, m, βH-5′). ^13^C NMR (CDCl_3_:CD_3_OD, 3:1) δ, ppm: 161.3, 150.2 × 2, 139.7, 131.5, 129.3, 128.0, 126.8 × 2, 121.8, 117.2 × 2, 73.9, 59.3, 50.2 × 2, 49.7 × 2, 39.5. HRMS, *m*/*z*: calculated for C_19_H_22_N_4_O_3_ [M + H]^+^ 355.1765 found 355.1764.

Compound **4b** was isolated by column chromatography on silica gel in CHCl_3_:MeOH (98:2) eluent system and then purified by PLC in CHCl_3_:MeOH (95:5). The target **4b** was obtained as yellowish transparent oil (66.7 mg, 0.15 mmol, 41%). Rf 0.4. ^1^H NMR (CDCl_3_) δ, ppm: 7.31–7.26 (2H, m, H-3‴, H-5‴), 7.04 (H, d, H-6), 6.98–6.88 (3H, m, H-2‴, H-6‴, H-4‴), 6.24–6.22 (1H, m, H-2′), 6.14–6.11 (1H, m, H-2⁗), 5.99–5.95 (1H, m H-3′), 5.86–5.82 (1H, m H-3⁗), 5.78–5.75 (1H, m, H-1′), 5.66–5.63 (1H, m, H-1⁗), 4.87–4.84 (1H, m, H-4′), 4.73 (1H, m H-4⁗), 3.33–3.29 (4H, m, 2H-3″, 2H-5″), 3.10–3.08 (4H, m, 2H-2″, 2H-6″), 2.87–2.73 (2H, m, αH-5′, αH-5⁗), 2.01–1.96 (1H, m, βH-5′), 1.67–1.60 (1H, m, βH-5⁗). ^13^C NMR (CDCl_3_) δ, ppm: 160.6, 150.8, 150.2, 139.5, 136.8, 132.1, 131.0, 129.3 × 2, 127.7, 125.2, 120.6, 116.6 × 2, 76.3, 74.4, 60.0, 56.4, 50.2 × 2, 49.5 × 2, 39.8, 37.4. HRMS, *m*/*z*: calculated for C_20_H_27_N_3_O_4_ [M + H]^+^ 374.2074; found 374.2072.

(±)-1-(4′-hydroxycyclopent-2′-en-1′-yl)-5-(3′,4′-dihydroisoquinolin-2′(1H)-yl)uracil **3c** and (±)-1-(4′-hydroxycyclopent-2′-en-1′-yl)-3-(4‴-hydroxycyclopent-2‴-en-1‴-yl)-5-(3″,4″-dihydroisoquinoline-2″(1H)-yl)uracil **4c** were synthesized from the starting 5-(3,4-dihydroisoquinolin-2(1H)-yl)uracil **2c**. Compound was **3c** isolated by column chromatography on silica gel in the system of eluents CHCl_3_:MeOH (98:2). The target **3c** was obtained as orange crystals (39.1 mg, 0.120 mmol, 29%). It was decomposed at 112 °C. Rf 0.29. ^1^H NMR (CDCl_3_:CD_3_OD, 3:1) δ, ppm: 7.13–7.01 (5H, m, H-5″, H-6″, H-7″, H-8″, H-6), 6.21–6.18 (1H, m, H-2′), 5.81–5.78 (1H, m, H-3′), 5.56–5.51 (1H, m, H-1′), 4.80–4.76 (1H, m, H-4′), 4.07 (2H, m, 1″-CH_2_), 3.27–3.23 (2H, m, 3″-CH_2_), 2.97–2.93 (2H, m, 4″-CH_2_), 2.87–2.77 (1H, m, αH-5′), 1.61–1.54 (1H, m, βH-5′). ^13^C NMR (CDCl_3_:CD_3_OD, 3:1): 161.4, 150.2, 139.7, 133.7, 133.5, 131.4, 128.7, 128.0, 126.8, 126.4, 126.3, 125.8, 73.9, 59.3, 52.5, 48.2, 39.6, 28.7. HRMS, *m*/*z*: calculated for C_18_H_19_N_3_O_3_ [M + H]^+^ 326.1499; found 326.1493.

Compound **4c** was isolated by column chromatography on silica gel in CHCl_3_:MeOH (99:1) eluent system and then purified by PLC in CHCl_3_:MeOH (95:5) system. The target **4c** was obtained as an orange oil (44.3 mg, 0.11 mmol, 26%). Rf 0.40. ^1^H NMR (CDCl_3_) δ, ppm: 7.17–7.10 (3H, m, H-8″, H-7″, H-6″), 7.07–7.02 (2H, H-5″, H-6), 6.23–6.19 (1H, m, H-2‴), 6.16–6.12 (1H, m, H-2′), 6.02–5.96 (1H, m, H-3‴), 5.85–5.81 (1H, m, H-3′), 5.79–5.76 (1H, m, H-1‴), 5.64–5.59 (1H, m, H-1′), 4.87–4.83 (1H, m, H-4‴), 4.76–4.71 (1H, m, H-4′), 4.11 (2H, s, 1″-CH_2_), 3.36–3.22 (2H, m, 3″-CH_2_), 2.98–2.95 (2H, m, 4″-CH_2_), 2.89–2.74 (2H, m, αH-5′, αH-5‴), 2.04–1.97 (1H, m, βH-5‴), 1.64–1.57 (1H, m, βH-5′). ^13^C NMR (CDCl_3_) δ, ppm: 160.3, 149.7, 138.8, 136.4, 133.4, 133.3, 131.6, 130.5, 128.3, 127.0, 125.9, 125.8, 125.3, 124.9, 75.8, 74.0, 59.6, 59.9, 52.3, 47.7, 39.3, 36.9, 28.2. HRMS, *m*/*z*: calculated for C_23_H_25_N_3_O_4_ [M + Na]^+^ 430.1737; found 430.1734.

(±)-1-(4′-Hydroxycyclopent-2′-en-1′-yl)-5-((4′-hexylphenyl)amino)uracil **3d** and (±)-1-(4′-hydroxycyclopent-2′-en-1′-yl)-3-(4‴-hydroxycyclopent-2‴-en-1‴-yl)-5-(-((4″-hexylphenyl)amino)uracil **4d** were synthesized from the starting 5-((4′-hexylphenyl)amino)uracil **2d**. Compound **3d** was isolated by column chromatography on silica gel in CHCl_3_:MeOH eluent system (98:2) and then purified by PLC in CHCl_3_:MeOH (95:5) to give an orange powder (38.1 mg, 0.10 mmol, 29.5%). M.p. 165.2 °C. Rf 0.32. ^1^H NMR (CDCl_3_) δ, ppm: 9.25 (1H, s, N^1^H), 7.54 (1H, s, H-6), 7.13–7.10 (2H, m, H-2″, H-6″), 7.03–7.01 (2H, m, H-3″, H-5″), 6.26–6.24 (1H, m, H-2′), 5.87–5.85 (1H, m, H-3′), 5.56–5.53 (1H, m, H-1′), 4.88–4.86 (1H, m, H-4′), 2.93–2.85 (1H, m, αH-5′), 2.58–2.53 (2H, m, α-CH_2_), 1.77–1.72 (1H, m, βH-5′), 1.64–1.54 (2H, m β-CH_2_), 1.31 (6H, s, (CH_2_)_3_), 0.93–0.88 (3H, m CH_3_). ^13^C NMR (CDCl_3_) δ, ppm: 160.5, 148.9, 139.5, 138.7, 131.8, 129.4×2, 128.1, 121.7, 119.7, 118.3 × 2, 74.8, 60.3, 39.7, 35.2, 31.7, 31.5, 29.0, 22.6, 14.1. HRMS, *m*/*z*: calculated for C_21_H_27_N_3_O_3_ [M + H]^+^ 370.2125; found 370.2123.

Compound **4d** was isolated by column chromatography on silica gel in CHCl_3_:MeOH eluent system (99:1), and then purified by PLC in CHCl_3_:MeOH (95:5) to give a red-orange oil (53 mg, 0.12 mmol, 34%). Rf 0.46. ^1^H NMR (CDCl_3_) δ, ppm: 7.38–7.37 (1H, m, H-6), 7.10–7.07 (2H, m, H-2″, H-6″), 6.93–6.90 (2H, m, H-3″, H-5″), 6.24–6.20 (1H, m, H-2′), 6.17–6.14 (1H, m, H-2′′′), 6.04–5.98 (1H, m, H-3′), 5.85–5.82 (1H, m, H-3‴), 5.80–5.77 (1H, m, H-1′), 5.68–5.61 (1H, m, H-1′′′), 4.88–4.84 (1H, m, H-4′), 4.76–4.72 (1H, m, H-4‴), 2.92–2.75 (2H, m, αH-5′, αH-5‴), 2.56–2.51 (2H, m, α-CH_2_), 2.04–1.98 (1H, m, βH-5‴), 1.71–1.64 (1H, m, βH-5′), 1.61–1.53 (2H, m, β-CH_2_), 1.35–1.28 (6H, m, (CH_2_)_3_), 0.92–0.87 (3H, m, CH_3_). ^13^C NMR (CDCl_3_) δ, ppm: 160.7, 149.3, 139.5, 137.2, 136.2, 132.2, 132.0, 130.7, 129.3 × 2, 128.5, 119.9, 118.2, 117.6 × 2, 76.2, 74.6, 60.4, 56.8, 39.9, 37.4, 35.2, 31.7, 31.5, 29.0, 22.6, 14.1. HRMS, *m*/*z*: calculated for C_26_H_33_N_3_O_4_ [M + Na]^+^ 474.2363; found 474.2361.

(±)-1-(4′-Hydroxycyclopent-2′-en-1′-yl)-5-((4″-tert-butyl)phenylamino)uracil **3e** and (±)-1-(1-(4′-hydroxycyclopent-2′-ene-1′-yl))-3-(4‴-hydroxycyclopent-2‴-en-1‴-yl)-5-((4″-tert-butyl)phenylamino)uracil **4e** were synthesized from the starting 5-((4′-tert-butylphenylamino)uracil **2e**. Compound **3e** was isolated by column chromatography on silica gel in CHCl_3_:MeOH (98:2) eluent system and then purified by PLC in CHCl_3_:MeOH (95:5) system. The target **3e** was obtained as a light yellow powder (71.2 mg, 0.17 mmol, 22%). M.p. 165 °C. Rf 0.30. ^1^H NMR (CDCl_3_) δ, ppm: 9.22 (1H, s, HN^1^), 7.46 (1H, s, H-6), 7.33–7.30 (2H, m, H-2″, H-6″), 7.01–6.99 (2H, m, H-3″, H-5″), 6.27–6.24 (1H, m, H-2′), 5.88–5.86 (1H, m, H-3′), 5.57–5.54 (1H, m, H-1′), 4.90–4.87 (1H, m, H-4′), 2.94–2.84 (1H, m, αH-5′), 1.79–1.71 (1H, m, βH-5′), 1.31 (9H, s, 3CH_3_). ^13^C NMR (CDCl_3_) δ, ppm: 160.6, 148.8, 144.0, 139.4, 138.7, 131.9, 126.4 × 2, 121.10, 119.8, 117.5 × 2, 74.8, 60.3, 39.7, 34.2, 31.4 × 3. HRMS, *m*/*z*: calculated for C_19_H_23_N_3_O_3_ [M + H]^+^ 342.1812; found 342.1809.

Compound **4e** was isolated by column chromatography on silica gel in CHCl_3_:MeOH eluent system (99:1), and then purified by PLC in CHCl_3_:MeOH (95:5) to give dark red oil (55.4 mg, 0.26 mmol, 24%). Rf 0.46. ^1^H NMR (CDCl_3_) δ, ppm: 7.40–7.39 (1H, d, H-6), 7.30–7.26 (2H, m, H-2″, H-6″), 6.95–6.91 (2H, m, H-3″, H-5″), 6.25–6.21 (1H, m, H-2‴), 6.16–6.13 (1H, m, H-2′), 6.03–5.98 (1H, m, H-3‴), 5.85–5.82 (1H, m, H-3′), 5.80–5.77 (1H, m, H-1‴), 5.67–5.62 (1H, m, H-1′), 4.88–4.84 (1H, m, H-4‴), 4.76–4.72 (1H, m, H-4′), 2.94–2.74 (2H, m, αH-5′, αH-5‴), 2.04–1.97 (1H, m, βH-5‴), 1.71–1.64 (1H, m, βH-5′), 1.30 (9H, s, 3CH_3_). ^13^C NMR (CDCl_3_) δ, ppm: 160.8, 149.3, 144.2, 139.5, 139.3, 137.1, 131.9, 130.8, 126.3 × 2, 119.7, 118.5, 116.9 × 2, 76.2, 74.6, 60.4, 56.8, 39.9, 37.4, 34.2, 31.4 × 3. HRMS, *m*/*z*: calculated for C_24_H_29_N_3_O_4_ [M + Na]^+^ 446.2050; found 446.2048.

(±)-4′-Hydroxycyclopent-2′-en-1′-yl)-5-((4′′-heptylphenyl)amino)uracil **3f** and (±)-1-(4′-hydroxycyclopent-2′-en-1′-yl)-3-(4‴-hydroxycyclopent-2‴-en-1‴-yl)-5-(-((4″-heptylphenyl)amino)uracil **4f** were synthesized from the starting 5-((4′-heptylphenyl)amino)uracil **2f**. Compound **3f** was isolated by column chromatography on silica gel in CHCl_3_:MeOH (98:2) eluent system. The target **3f** was obtained as a light brown powder (30.5 mg, 0.08 mmol, 24%) Decomposed at 192 °C. Rf 0.29. ^1^H NMR (CDCl_3_:CD_3_OD, 3:1) δ, ppm: 7.44 (1H, s, H-6), 7.06–7.03 (2H, m, H-2″, H-6″), 6.88–6.83 (2H, m, H-3″, H-5″), 6.20–6.17 (1H, m, H-2′), 5.81–5.78 (1H, m, H-3′), 5.58–5.54 (1H, m, H-1′), 4.79–4.75 (1H, m, H-4′), 2.90–2.80 (1H, m, αH-5′), 2.52–2.47 (2H, m, α-CH_2_), 1.62–1.52 (3H, m, βH-5′, β-CH_2_), 1.29–1.24 (8H, m, (CH_2_)_4_), 0.88–0.83 (3H, m, CH_3_). ^13^C NMR (CDCl_3_:CD_3_OD, 3:1) δ, ppm: 160.4, 151.5, 139.7, 135.5, 131.2, 129.2 × 2, 129.1, 121.0, 118.7, 116.9×2, 74.0, 59.3, 39.7, 35.0, 31.7, 31.5, 29.1, 22.5, 13.8. HRMS, *m*/*z*: calculated for C_22_H_29_N_3_O_3_ [M + H]^+^ 384.2282; found 384.2272.

Compound **4f** was isolated by column chromatography on silica gel in CHCl_3_:MeOH eluent system (99:1), and then purified by PLC in CHCl_3_:MeOH (95:5) to give a brown oil (33.6 mg, 0.07, mmol, 21%). Rf 0.38. ^1^H NMR (CDCl_3_) δ, ppm: 7.41–7.40 (1H, d, H-6), 7.14–7.08 (2H, m, H-2″, H-6″), 6.99–6.93 (2H, m, H-3″, H-5″), 6.24–6.22 (1H, m, H-2‴), 6.18–6.15 (1H, m, H-2′), 6.04–5.98 (1H, m, H-3‴), 5.85–5.83 (1H, m, H-3′), 5.80–5.78 (1H, m, H-1‴), 5.66–5.64 (1H, m, H-1′), 4.88–4.85 (1H, m, H-4‴), 4.76–4.74 (1H, m, H-4′), 2.93–2.76 (2H, m, αH-5‴, αH-5′), 2.57–2.52 (2H, m, α-CH_2_), 2.04–1.98 (1H, d, βH-5‴), 1.72–1.65 (1H, m, βH-5′), 1.61–1.54 (2H, m, β-CH_2_), 1.32–1.28 (8H, m, (CH_2_)_4_), 0.92–0.88 (3H, m, CH_3_). ^13^C NMR (CDCl_3_) δ, ppm: 160.7, 149.3, 139.4, 139.3, 137.3, 132.1, 130.7, 129.4 × 2, 128.7, 119.8, 118.4, 117.8 × 2, 76.2, 74.7, 60.4, 56.8, 39.9, 37.5, 35.2, 31.8, 31.6, 29.7, 29.3, 22.7, 14.1. HRMS, *m*/*z*: calculated for C_27_H_35_N_3_O_4_ [M + H]^+^ 466.2700; found 466.2690.

(±)-1-(4′-Hydroxycyclopent-2′-en-1′-yl)-5-((4″-isopropylphenyl)amino)uracil **3g** and (±)-1-(4′-hydroxycyclopent-2′-en-1′-yl)-3-(4‴-hydroxycyclopent-2‴-en-1‴-yl)-5-((4″-isopropylphenyl)amine)uracil **4g** were synthesized from the starting 5-((4′-iso-propylphenyl)amino)uracil **2g**. Compound **3g** was isolated by column chromatography on silica gel in CHCl_3_:MeOH (98:2) eluent system. The target **3g** was obtained as a light orange powder (47 mg, 0.14 mmol, 35%). Decomposed at 201 °C. Rf 0.27. 1H NMR (CDCl_3_:CD_3_OD, 3:1) δ, ppm: 7.41 (1H, s, H-6), 7.12–7.08 (2H, m, H-2″, H-6″), 6.91–6.86 (2H, m, H-3″, H-5″), 6.21–6.17 (1H, m, H-2′), 5.81–5.78 (1H, m, H-3′), 5.58–5.53 (1H, m, H-1′), 4.79–4.75 (1H, m, H-4′), 2.90–2.77 (2H, m, αH-5′, -CH-), 1.62–1.54 (1H, m, βH-5′), 1.20–1.17 (6H, d, 2CH_3_). ^13^C NMR (CDCl_3_:CD_3_OD, 3:1) δ 161.5, 149.5, 141.6, 140.0, 139.7, 131.2, 127.2 × 2, 121.1, 120.3, 116.9 × 2, 74.0, 59.4, 39.8, 33.3, 23.8 × 2. HRMS, *m*/*z*: calculated for C_18_H_21_N_3_O_3_ [M + H]^+^ 328.1656; found 328.1652.

Compound **4g** was isolated by column chromatography on silica gel in CHCl_3_:MeOH eluent system (99:1), and then purified by PLC in CHCl_3_:MeOH (95:5) to give a red-orange oil (44.5 mg, 0.11 mmol, 27%). Rf 0.4. ^1^H NMR (CDCl_3_) δ, ppm: 7.34–7.33 (1H, d, H-6), 7.17–7.12 (2H, m, H-2″, H-6″), 6.95–6.90 (2H, m, H-3″, H-5″), 6.24–6.21 (1H, m, H-2‴), 6.19–6.15 (1H, m, H-2′), 6.05–5.99 (2H, m, H-3‴, 5-NH), 5.87–5.84 (1H, m, H-3′), 5.81–5.78 (1H, m, H-1‴, 5.69–5.62 (1H, m, H-1′), 4.90–4.86 (1H, m, H-4‴), 4.76–4.74 (1H, d, H-4′), 2.96–2.79 (3H, m, αH-5‴, αH-5′, -CH-), 2.05–1.98 (1H, d, βH-5‴), 1.72–1.65 (1H, m, βH-5′), 1.25–1.22 (6H, m, 2CH_3_). ^13^C NMR (CDCl_3_) δ, ppm: 160.8, 149.2, 141.9, 139.6, 139.6, 137.3, 132.2, 130.8, 127.4 × 2, 120.0, 117.5, 117.4 × 2, 76.2, 74.7, 60.4, 56.8, 39.8, 37.5, 33.4, 24.1 × 2. HRMS, *m*/*z*: calculated for C_23_H_27_N_3_O_4_ [M + H]^+^ 410.2074; found 410.2066.

(±)-1-(4′-Hydroxycyclopent-2′-en-1′-yl)-5-((4″-(hexyloxy)phenyl)amino)uracil **3h** and (±)-1-(4′-hydroxycyclopent-2′-en-1′-yl)-3-(4‴-hydroxycyclopent-2‴-en-1‴-yl)-5-((4″-(hexyloxy) phenyl)amino)uracil **4h** were synthesized from the starting 5-((4′-hexyloxyphenyl)amino)uracil **2h**. Compound **3h** was isolated by column chromatography on silica gel in CHCl_3_:MeOH (98:2) eluent system. The target **3h** was obtained as dark brown crystals (25 mg, 0.06 mmol, 20%). M.p. 175 °C. Rf 0.34. ^1^H NMR (CDCl_3_:CD_3_OD, 3:1) δ, ppm: 7.21 (1H, s, H-6), 6.95–6.90 (2H, d, H-2″, H-6″), 6.84–6.78 (2H, m, H-3″, H-5″), 6.17–6.14 (1H, m, H-2′), 5.78–5.75 (1H, m, H-3′), 5.55–5.48 (1H, m, H-1′), 4.77–4.72 (1H, m, H-4′), 3.91–3.87 (2H, m, α-CH_2_), 2.87–2.77 (1H, m, αH-5′), 1.77–1.68 (2H, m, β-CH_2_), 1.60–1.52 (1H, m, βH-5′), 1.47–1.38 (2H, m, γ-CH_2_), 1.35–1.28 (4H, m, (CH_2_)_2_), 0.90–0.86 (3H, m, CH_3_). ^13^C NMR (CDCl_3_:CD_3_OD, 3:1) δ, ppm: 161.1, 154.7, 149.5, 139.8, 134.9, 131.1, 123.9, 120.7, 120.1×2, 115.5×2, 74.0, 68.6, 59.4, 39.7, 31.5, 29.2, 25.6, 22.5, 13.7. HRMS, *m*/*z*: calculated for C_21_H_27_N_3_O_4_ [M + H]^+^ 386.2074; found 386.2065.

Compound **4h** was isolated by column chromatography on silica gel in a CHCl_3_:MeOH eluent system (99:1), and then purified by PLC in a CHCl_3_:MeOH (95:5) to give dark red oil (47 mg, 0.11 mmol, 30%). Rf 0.42. ^1^H NMR (CDCl_3_) δ, ppm: 7.19–7.18(1H, d, H-6), 6.97–6.94 (2H, m, H-2″, H-6″), 6.86–6.82 (2H, m, H-3″, H-5″), 6.20–6.14 (2H, m, H-2‴, H-2′), 6.04–5.98 (1H, m, H-3‴), 5.83–5.77 (2H, m, H-3′, H-1‴), 5.64–5.58 (1H, m, H-1′), 4.85–4.82 (1H, d, H-4‴), 4.76–4.72 (1H, m, H-4′), 3.93–3.89 (2H, m, α-CH_2_), 2.90–2.75 (2H, m, αH-5‴, αH-5′), 2.03–1.98(1H, d, βH-5‴), 1.81–1.72 (2H, m, β-CH_2_), 1.68–1.61 (1H, m, βH-5′), 1.50–1.43 (2H, m, γ-CH_2_), 1.37–1.32 (4H, m, (CH_2_)_2_), 0.94–0.89 (3H, m, CH_3_). ^13^C NMR (CDCl_3_) δ, ppm: 160.6, 154.6, 149.2, 139.3, 137.2, 134.6, 132.0, 130.8, 121.3, 120.5 × 2, 116.1, 115.5 × 2, 76.2, 74.6, 68.5, 60.4, 56.8, 39.8, 37.5, 31.6, 29.3, 25.7, 22.6, 14.0. HRMS, *m*/*z*: calculated for C_26_H_33_N_3_O_5_ [M + Na]^+^ 490.2312; found 490.2303.

(±)-1-(4′-Hydroxycyclopent-2′-en-1′-yl)-5-((4″-(heptyloxy)phenyl)amino)uracil **3i** and (±)-1-(4′-hydroxycyclopent-2′-en-1′-yl)-3-(4‴-hydroxycyclopent-2‴-en-1‴-yl)-5-((4″-(heptyloxy)phenyl)amino)uracil **4i** were synthesized from the starting 5-((4′-heptyloxyphenyl)amino)uracil **2i**. Compound **3i** was isolated by column chromatography on silica gel in CHCl_3_:MeOH (98:2) eluent system. The target **3i** was obtained as small dark-brown crystals (18 mg, 0.045 mmol, 14%). M.p. 162 °C. Rf 0.27. ^1^H-NMR (CDCl_3_:CD_3_OD, 3:1) δ, ppm: 7.20 (1H, s, H-6), 6.92–6.89 (2H, m, H-2″, H-6″), 6.83–6.79 (2H, m, H-3″, H-5″), 6.17–6.14 (1H, m, H-2′), 5.78–5.75 (1H, m, H-3′), 5.56–5.50 (1H, m, H-1′), 4.77–4.72 (1H, m, H-4′), 3.91–3.87 (2H, m, α-CH_2_), 2.87–2.77 (1H, m, αH-5′), 1.77–1.68 (2H, m, β-CH_2_), 1.59–1.51 (1H, m, βH-5′), 1.46–1.37 (2H, m, γ-CH_2_), 1.35–1.27 (6H, m, (CH_2_)_3_), 0.89–0.84 (3H, m, CH_3_). ^13^C NMR (CDCl_3_:CD_3_OD, 3:1) δ, ppm: 160.8, 153.6, 148.9, 139.1, 134.8, 130.7, 121.0, 119.1 × 2, 118.5, 115.0 × 2, 73.4, 68.1, 58.8, 39.2, 31.2, 28.7, 28.4, 25.4, 21.9, 13.2. HRMS, *m*/*z*: calculated for C_22_H_29_N_3_O_4_ [M + H]^+^ 400.2231; found 400.2221.

Compound **4i** was isolated by column chromatography on silica gel in CHCl_3_:MeOH eluent system (99:1), and then purified by PLC in CHCl_3_:MeOH (95:5) to give a dark red oil (48 mg, 0.10 mmol, 31%). Rf 0.43. ^1^H-NMR (CDCl_3_) δ, ppm: 7.17–7.16 (1H, d, H-6), 6.98–6.93 (2H, m, H-2″, H-6″), 6.87–6.81 (2H, m, H-3″, H-5″), 6.20–6.15 (2H, m, H-3‴, H-3‴), 6.04–5.99 (1H, m, H-2‴), 5.84–5.77 (2H, m, H-2′, H-1‴), 5.64–5.58 (1H, m, H-1′), 4.87–4.83 (1H, m, H-4‴), 4.76–4.72 (1H, m, H-4′), 3.94–3.89 (2H, m, α-CH_2_), 2.91–2.76 (2H, m, αH-5‴, αH-5′), 2.04–1.98 (1H, d, βH-5‴), 1.81–1.72 (2H, m, β-CH_2_), 1.69–1.62 (1H, m, βH-5′), 1.51–1.39 (2H, m, γ-CH_2_), 1.38–1.30 (6H, m, (CH_2_)_3_), 0.94–0.89 (3H, m, CH_3_). ^13^C NMR (CDCl_3_) δ, ppm: 160.1, 154.1, 148.7, 138.6, 136.8, 134.0, 131.7, 130.2, 120.8, 120.1 × 2, 115.3, 115.0 × 2, 75.7, 74.2, 67.9, 59.9, 56.3, 39.3, 37.0, 31.3, 28.8, 28.5, 25.5, 22.1, 13.5. HRMS, *m*/*z*: calculated for C_27_H_35_N_3_O_5_ [M + Na]^+^ 504.2469; found 504.2461

(±)-1-(4′-Hydroxycyclopent-2′-en-1′-yl)-5-(3″,5″-dimethylphenoxy)uracil **10a** and (±)-1-(4′-hydroxycyclopent-2′-en-1′-yl)-3-(4‴-hydroxycyclopent-2‴-en-1‴-yl)-5-(3″,5″-dimethylphenoxy)uracil **11a** were synthesized from the starting 5-(3,5-dimethylphenoxy)uracil **5a**. Compound **10a** was isolated by column chromatography on silica gel in CHCl_3_:MeOH (98:2) eluent system. The target **10a** was obtained as a light yellow powder (40 mg, 0.13 mmol, 30%). 188.9 °C. Rf 0.37. ^1^H-NMR (CDCl_3_:CD_3_OD, 3:1) δ, ppm: 7.49 (1H, s, H-6), 6.65 (1H, s, H-4″), 6.52 (2H, s, H-2″, H-6″), 6.19–6.16 (1H, m, H-2′), 5.80–5.77 (1H, m, H-3′), 5.55–5.50 (1H, m, H-1′), 4.77–4.72 (1H, m, H-4′), 2.94–2.84 (1H, m, αH-5′), 2.24 (6H, s, 2CH_3_), 1.57–1.49 (1H, m, βH-5′). ^13^C NMR (CDCl_3_:CD_3_OD, 3:1) δ, ppm: 159.9, 157.5, 150.3, 140.2 × 2, 139.4, 132.4, 131.5, 130.8, 124.7, 113.2 × 2, 74.0, 59.6, 40.1, 21.2 × 2. HRMS, *m*/*z*: calculated for C_17_H_18_N_2_O_4_ [M + H]^+^ 315.1339; found 315.1337.

Compound **11a** was isolated by column chromatography on silica gel in CHCl_3_:MeOH eluent system (99:1) to give a yellowish oil (39 mg, 0.1 mmol, 23%). Rf 0.42. ^1^H-NMR (CDCl_3_) δ, ppm: 7.46 (1H, s, H-6), 6.70 (1H, s, H-4′′), 6.52 (2H, s, H-2″, H-6″), 6.23–6.19 (1H, m, H-2‴), 6.15–6.10 (1H, m, H-2′), 5.95–5.92 (1H, m, H-3‴), 5.85–5.81 (1H, m, H-3′), 5.79–5.75 (1H, m, H-1‴), 5.65–5.59 (1H, m, H-1′), 4.85–4.80 (1H, m, H-4‴), 4.73–4.70 (1H, d, H-4′), 2.94–2.74 (2H, m, αH-5‴, αH-5′), 2.28 (6H, s, 2CH_3_), 2.03–1.96 (1H, m, βH-5‴), 1.67–1.58 (1H, m, βH-5′). ^13^C NMR (CDCl_3_) δ, ppm: 159.5, 157.4, 150.4, 140.1, 139.5 × 2, 137.3, 131.4, 130.7, 130.6, 130.5, 124.9, 113.4 × 2, 76.2, 74.5, 60.3, 56.7, 40.2, 37.4, 21.4 × 2. HRMS, *m*/*z*: calculated for C_22_H_24_N_2_O_5_ [M + Na]^+^ 419.1577; found 419.1569.

(±)-1-(4′-Hydroxycyclopent-2′-en-1′-yl)-5-(4″-butoxyphenoxy)uracil **10b** and (±)-1-(4′-hydroxycyclopent-2′-en-1′-yl)-3-(4‴-hydroxycyclopent-2‴-en-1‴-yl)-5-(4″-butoxyphenoxy) uracil **11b** were synthesized from the starting 5-(4-chlorophenoxy)uracil 5b. Compound **10b** was isolated by column chromatography on silica gel in CHCl_3_:MeOH (98:2) eluent system. The target **10b** was obtained as a light yellow powder (23 mg, 0.06 mmol, 18%). M.p. 174.8 °C. Rf 0.31. ^1^H-NMR (CDCl_3_:CD_3_OD, 3:1) δ, ppm: 7.46 (1H, s, H-6), 6.90–6.84 (2H, m, H-2″, H-6″), 6.82–6.78 (2H, m, H-3″, H-5″), 6.19–6.15 (1H, m, H-2′), 5.78–5.75 (1H, m, H-3′), 5.54–5.49 (1H, m, H-1′), 4.76–4.72 (1H, m, H-4′), 3.91–3.87 (2H, m, α-CH_2_), 2.92–2.81 (1H, m, αH-5′), 1.76–1.67 (2H, m, β-CH_2_), 1.55–1.52 (1H, m, βH-5′), 1.50–1.39 (2H, m, γ-CH_2_), 0.97–0.92 (3H, m, CH_3_). ^13^C NMR (CDCl_3_:CD_3_OD, 3:1) δ, ppm: 160.3, 155.0, 151.2, 150.4, 140.2, 132.6, 131.6, 130.8, 117.0 × 2, 115.4 × 2, 73.9, 68.3, 59.4, 40.0, 31.2, 19.1, 13.6. HRMS, *m*/*z*: calculated for C_19_H_22_N_2_O_5_ [M + H]^+^ 359.1601; found 359.1599.

Compound **11b** was isolated by column chromatography on silica gel in CHCl_3_:MeOH eluent system (99:1) to give a yellowish oil (48.5 mg, 0.11 mmol, 31%). Rf 0.47. ^1^H-NMR (CDCl_3_) δ, ppm: 7.41 (1H, s, H-6), 6.89–6.78 (4H, m, H-2″, H-6″, H-3″, H-5″), 6.20–6.16 (1H, m, H-2‴), 6.13–6.09 (1H, m, H-2′), 5.95–5.89 (1H, m, H-3‴), 5.81–5.74 (2H, m, H-3′, H-1‴), 5.62–5.56 (1H, m, H-1′), 4.82–4.76 (1H, m, H-4‴), 4.71–4.69 (1H, d, H-4′), 3.94–3.89 (2H, m, α-CH_2_), 2.90–2.72 (2H, m, αH-5‴, αH-5′), 2.00–1.95 (1H, m, βH-5‴), 1.79–1.70 (2H, m, β-CH_2_), 1.64–1.56 (1H, m, βH-5′), 1.52–1.42 (2H, m, γ-CH_2_), 1.00–0.95 (3H, m, CH_3_). ^13^C NMR (CDCl_3_) δ, ppm: 159.5, 155.2, 150.8, 150.3, 140.1, 137.2, 132.1, 131.3, 130.7, 129.7, 117.3 × 2, 115.6 × 2, 76.2, 74.4, 68.2, 60.3, 56.7, 40.1, 37.3, 31.4, 19.2, 13.8. HRMS, *m*/*z*: calculated for C_24_H_28_N_2_O_6_ [M + H]^+^ 441.2020; found 441.2016.

(±)-1-(4′-Hydroxycyclopent-2′-en-1′-yl)-5-(4″-chlorophenoxy)uracil **10c** and (±)-1-(4′-hydroxycyclopent-2′-en-1′-yl)-3-(4‴-hydroxycyclopent-2‴-en-1‴-yl)-5-(4″-chlorophenoxy)uracil **11c** were synthesized from the starting 5-(4-butoxyphenoxy)uracil 5c. Compound **10c** was isolated by column chromatography on silica gel in CHCl_3_:MeOH (98:2) eluent system. The target **10c** was obtained as a pale yellow powder (58 mg, 0.18 mmol, 43%). M.p. 190 °C. Rf 0.29. ^1^H NMR (CDCl_3_:CD_3_OD, 3:1) δ, ppm: 7.57 (1H, s, H-6), 7.24–7.18 (2H, m, H-2″, H-6″), 6.89–6.84 (2H, m, H-3″, H-5″), 6.20–6.17 (1H, m, H-2′), 5.79–5.77 (1H, m, H-3′), 5.56–5.51 (1H, m, H-1′), 4.77–4.72 (1H, m, H-4′), 2.92–2.82 (1H, m, αH-5′), 1.56–1.49 (1H, m, βH-5′). ^13^C NMR (CDCl_3_:CD_3_OD, 3:1) δ, ppm: 160.0, 156.2, 150.4, 140.3, 133.1, 131.1, 130.7, 129.4 × 2, 127.8, 116.9 × 2, 73.9, 59.5, 40.0 HRMS, *m*/*z*: calculated for C_15_H_13_ClN_2_O_4_ [M + H]^+^ 321.0637; found 321.0643.

Compound **11c** was isolated by column chromatography on silica gel in CHCl_3_:MeOH eluent system (99:1) to give a yellowish oil (58 mg, 0.14 mmol, 34%). Rf 0.38. ^1^H NMR (CDCl_3_) δ, ppm: 7.54 (1H, s, H-6), 7.27–7.24 (2H, m, H-2″, H-6″), 6.88–6.84 (2H, m, H-3″, H-5″), 6.26–6.22 (1H, m, H-2‴), 6.16–6.12 (1H, m, H-2′), 5.95–5.90 (1H, m, H-3‴), 5.85–5.83 (1H, m, H-3′), 5.78–5.74 (1H, m, H-1‴), 5.65–5.63 (1H, m, H-1′), 4.85–4.83 (1H, m, H-4‴), 4.73–4.71 (1H, d, H-4‴), 2.92–2.74 (2H, m, αH-5‴, αH-5′), 2.00–1.95 (1H, d, βH-5‴), 1.67–1.58 (1H, m, βH-5′). ^13^C NMR (CDCl_3_) δ, ppm: 158.7, 155.5, 149.8, 139.7, 136.8, 130.8, 130.6, 130.1, 129.9, 129.1 × 2, 127.6, 116.5 × 2, 75.6, 73.8, 59.8, 56.3, 39.6, 36.8. HRMS, *m*/*z*: calculated for C_20_H_19_ClN_2_O_5_ [M + Na]^+^ 425.0875; found 425.0867.

(±)-1-(4′-Hydroxycyclopent-2′-en-1′-yl)-5-(4″-(tert-butyl)phenoxy)uracil **10d** and (±)-1-(4′-hydroxycyclopent-2′-en-1′-yl)-3-(4‴-hydroxycyclopent-2‴-en-1‴-yl)-5-(4″-(tert-butyl) phenoxy)uracil **11d** were synthesized from the starting 5-(4-tert-butylphenoxy)uracil **5d**. Compound **10d** was isolated by column chromatography on silica gel in CHCl_3_:MeOH (98:2) eluent system. The target **10d** was obtained as a pale-yellow powder (68.6 mg, 0.2 mmol, 53%). M.p. 200 °C. Rf 0.27. ^1^H NMR (CDCl_3_:CD_3_OD, 3:1) δ, ppm: 7.52 (1H, s, H-6), 7.29–7.24 (2H, m, H-2″, H-6″), 6.86–6.81 (2H, m, H-3″, H-5″), 6.19–6.16 (1H, m, H-2′), 5.79–5.76 (1H, m, H-3′), 5.56–5.50 (1H, m, H-1′), 4.77–4.72 (1H, m, H-4′), 2.93–2.83 (1H, m, αH-5′), 1.56–1.48 (1H, m, βH-5′), 1.25 (9H, s, 3CH_3_). ^13^C NMR (CDCl_3_:CD_3_OD, 3:1) δ, ppm: 160.2, 155.3, 150.4, 145.7, 140.2, 132.67, 131.5, 130.7, 126.3 × 2, 114.9 × 2, 73.9, 59.4, 40.0, 34.1, 31.2 × 3. HRMS, *m*/*z*: calculated for C_19_H_22_N_2_O_4_ [M + H]^+^ 343.1652; found 343.1652.

Compound **11d** was isolated by column chromatography on silica gel in CHCl_3_:MeOH eluent system (99:1) to give a yellow-green oil (69 mg, 0.13 mmol, 34%). Rf 0.38. ^1^H NMR (CDCl_3_) δ, ppm: 7.47 (1H, s, H-6), 7.34–7.29 (2H, m, H-2″, H-6″), 6.85–6.80 (2H, m, H-3″, H-5″), 6.22–6.17 (1H, m, H-2‴), 6.15–6.10 (1H, m, H-2′), 5.97–5.91 (1H, m, H-3‴), 5.83–5.75 (2H, m, H-3′, H-1‴), 5.65–5.59 (1H, m, H-1′), 4.83–4.78 (1H, m, H-4‴), 4.73–4.69 (1H, m, H-4′), 2.91–2.74 (2H, m, αH-5‴, αH-5′), 2.03–1.95 (1H, m, βH-5‴), 1.67–1.57 (1H, m, βH-5′), 1.31 (9H, s, 3CH_3_). ^13^C NMR (CDCl_3_) δ, ppm: 159.6, 155.2, 150.4, 145.8, 140.1, 137.3, 131.4, 130.9, 130.8, 130.6, 126.5 × 2, 115.0 × 2, 76.2, 74.45, 60.3, 56.8, 40.1, 37.4, 34.2, 31.5 × 3. HRMS, *m*/*z*: calculated for C_24_H_28_N_2_O_5_ [M + Na]^+^ 447.1890; found 447.1879.

### 3.2. Biology

Cells and viruses. The compounds were evaluated against the following viruses: respiratory syncytial virus (RSV) strain A Long, Influenza virus A (subtypes H1N1, H3N2), influenza virus B, yellow fever virus (YFV) strain 17 D, coronaviruses (HcoV-OC43, HcoV-229E, NL63), herpes simplex virus type 1 (HSV-1) strain KOS, varicella-zoster virus (VZV) strains TK^+^ and TK^−^, human cytomegalovirus (HCMV) strains AD-169 and Davis. Cell lines used: human embryonic lung fibroblasts (HEL 299), human liver carcinoma (Huh 7) and Madin-Darby canine kidney (MDCK) were obtained from American Type Culture Collection (ATCC).

Cell toxicity evaluation in HEL299, Huh 7 and MDCK cells. Cells were seeded at a rate of 5 × 10^3^ cells/well into 96-well plates and allowed to proliferate for 24 h. Then, medium containing different concentrations of the test compounds starting at 100 µM was added. After three days of incubation at 37 °C, the cell number was determined with a Beckman Coulter counter. The cytostatic concentrations or CC_50_ (compound concentration required reducing cell proliferation by 50%) were estimated from graphic plots of the number of cells (percentage of control) as a function of the concentration of the test compounds.

With regard to the cytopathicity or plaque reduction test, confluent cell cultures in 96-well plates were inoculated with 100 CCID_50_ of virus (CCID_50_ is a virus dose to infect 50% of the cell cultures) or with 20 plaque forming units (PFU). Following a 2 h adsorption period, viral inoculum was removed and the cell cultures were incubated in the presence of varying concentrations of the test compounds starting at 100 µM (solution in DMSO). Viral cytopathicity or plaque formation was recorded as soon as it reached completion in the control virus-infected cell cultures that were not treated with the test compounds. Antiviral activity was expressed as the EC_50_ (compound concentration required reducing virus-induced cytopathicity or viral plaque formation by 50%).

## Data Availability

This research was funded by Russian Foundation for Basic Research, grant number 18-29-08010 (design and synthesis) and Russian Science Foundation, grant number 19-74-10048 (characterization of target products).

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
