# Peer review of "New Derivatives of 5-Substituted Uracils: Potential Agents with a Wide Spectrum of Biological Activity"

_molecules, 2022, doi:10.3390/molecules27092866_

Round 1
Reviewer 1 Report
The paper “New derivatives of 5-substituted uracils: potential agents with a wide spectrum of biological activity” by A. L. Khandazhinskaya describes the synthesis and primary screening of the biological activity of some new nucleoside analogs.
The paper is well written, and the purpose of the study is clear, however the chemistry underlying the synthesis of the compounds suffers from novelty, using only procedures already reported in the literature for similar substrates, as the same authors report.
Furthermore, the synthesis leads to the preparation of the compounds in a racemic mixture without a description or comments on the relative stereochemistry of the obtained compounds. Furthermore, the choice to test this mixture, in my opinion, does not provide the necessary information on the real effectiveness of these compounds which could be even higher than that observed due to the presence of a presumably inactive enantiomer, nor which of the two enantiomers of the mixture is endowed with biological activity.
Nonetheless, I believe the paper may be published in molecules after major reviews below.
Even if Trost reaction is well-known, a discussion about the stereochemistry outcome of the reaction should be added, pointing out that the final compounds are racemic mixtures.
Line 172-174 I don't understand what the phrase “It should be noted that the use of potassium tert-butoxide (tBuOK) as a base in the third and fourth stages significantly reduces the yield of compounds 5a-d” refers to. Please clarify.
Scheme 1 and 2. The relative stereochemistry of nucleosides should be added and the obtaining the racemic mixture should be clearly indicated in the scheme, in the experimental part and in the supporting information.
Scheme 2 do not show that compounds 7a-d, 8a-d and 9a-d are synthetic intermediates and their isolation and characterization is not reported in the experimental part. The full characterization of at least two of them should be added. Of an entire synthetic scheme there are only two characterized compounds, and the absence of the other intermediates is not justified.
The choice to test a racemic mixture should be motivated by the authors.
In the experimental section:
ml should be replaced by mL
mp values should be accompanied by the recrystallization solvent
In the supporting information the 1H NMR and 13C NMR spectra should be shown as low as 0 ppm as evidence of actual purity
Author Response
The authors are grateful to the Reviewer for constructive criticism. We appreciate all the reviewer’s comments and added the required data. It made the paper stronger and we hope that now it is appropriate for the publication.
Even if Trost reaction is well-known, a discussion about the stereochemistry outcome of the reaction should be added, pointing out that the final compounds are racemic mixtures.
Thank you for the comment. The discussion about the stereochemistry outcome of the reaction is added in the text.
Line 172-174 I don't understand what the phrase “It should be noted that the use of potassium tert-butoxide (tBuOK) as a base in the third and fourth stages significantly reduces the yield of compounds 5a-d” refers to. Please clarify.
We have clarified in the paper text: “It should be noted that the use of potassium tert-butoxide (tBuOK) instead of NaH reduces the yield of compounds 5a-d to 17%.”
Scheme 1 and 2. The relative stereochemistry of nucleosides should be added and the obtaining the racemic mixture should be clearly indicated in the scheme, in the experimental part and in the supporting information.
Done
Scheme 2 do not show that compounds 7a-d, 8a-d and 9a-d are synthetic intermediates and their isolation and characterization is not reported in the experimental part. The full characterization of at least two of them should be added. Of an entire synthetic scheme there are only two characterized compounds, and the absence of the other intermediates is not justified.
The authors apologize for the inaccuracy in Scheme 2. Compounds 7a-d and 8a-d are synthetic intermediates; they were not isolated and characterized. We now have shown it in the scheme 2 and added data on compounds 9a-c in the experimental part
The choice to test a racemic mixture should be motivated by the authors.
The use of 6-oxobicyclo[3.1.0]hex-2-ene in the Trost condensation affords racemic products, which can be used for primary activity screening. It is common practice, so it helps to save time and money. Individual enantiomers of the most active compounds can be latter catalytically cleaved using various lipases [Amat, M.; Coll, M. D.; Bosch, J.; Espinosa, E.; Molins, E. Tetrahedron: Asymmetry 1997, 8, 935. Merlo, V.; Reece, F. J.; Roberts, S. M.; Gregson, M.; Storer, R. J. Chem. Perkin Trans. 1 1993, 1717. ] for better understanding of each enantiomer’s contribution to the activity. The work described in this paper gave us few compounds (3f, 3i, 4d, 4e, 11a and 11d) for which we are going to get individual enantiomers, test their activity and the results will be published as soon as they become available.
In the experimental section:
ml should be replaced by mL
Done
mp values should be accompanied by the recrystallization solvent
Done. Missing recrystallization solvents added to the general procedure description
In the supporting information the 1H NMR and 13C NMR spectra should be shown as low as 0 ppm as evidence of actual purity
Done
Reviewer 2 Report
The manuscript molecules-1674577 by Vasily A et al presents the synthesis, characterization and antival activity of new derivatives of 5-substituted uracils.
As comments/suggestions:
In my opinion, the study of biological activity should be presented in more detail.
In which solvent was the dissolution of the compounds performed in order to test the antiviral activity?
How is the difference in biological activity and toxicity between the compounds explained?
A possible antiviral mechanism of action for synthesized compounds?
Author Response
The authors appreciate the Reviewer’s questions and comments which helped to make the paper better and hope that now it is appropriate for the publication.
In which solvent was the dissolution of the compounds performed in order to test the antiviral activity?
Missing solvent (DMSO) is added to the experimental part
How is the difference in biological activity and toxicity between the compounds explained?
According to our data several factors have influence on the antiviral activity and toxicity of the compounds. 5-Arylaminouracils and 5-phenyloxyuracils are non-toxic and inactive, i.e. the presence of a substituent at N1 is critical. No less important the nature of the substituent at 5 position of uracil, which has different effect on the properties of N1-monocyclopentene and N1,N3-biscyclopentene derivatives. Thus, to exhibit activity against HCov, the monocyclopentene derivative needs a heptyl or oxyheptyl residue in the para-position of the aryl fragment, while the biscyclopentene derivative needs a hexyl residue. The 4-tBu substituent results in toxicity for both the monocyclopentene and the biscyclopentene derivative. The only compound with toxicity among the monocyclopentene derivatives of 5-phenyloxyuracil, that did not show any antiviral activity, was an analog with a 3,5-dimethylphenyloxy moiety.
The fact that none of the compounds showed any toxicity for the cells where activity was found is for different mechanisms/targets for activity and toxicity. We plan to continue work for better understanding SAR and the mechanism of action of 5'-substituted derivatives of 5'-norcarbocyclic uridine analogues.
A possible antiviral mechanism of action for synthesized compounds?
5'-Norcarbocyclic analogs of 2',3'-dideoxy-2',3'-didehydrouridine with different substituents in the 5 position of the heterocyclic base have activity against various viral and bacterial pathogens. Being 5'-norcarbocyclic analogs these compounds are definitely not substrates of kinases and cannot act as nucleoside inhibitors (convert to the corresponding triphosphates and be terminal substrates of RNA polymerases). The possible way in case of RNA viruses is inhibition of RNA polymerases by nonnucleoside mechanism. This hypothesis should be supported by additional experiments, which are in our plans but will take time.
Round 2
Reviewer 1 Report
all the suggested improvements have been reported by the authors and I suggest the publication of the paper in the present form. Only one final detail should be added by the authors prior to publication, with no need for me to review the manuscript.
The relative stereochemistry of the products obtained from the Trost reaction, specifying that they are cis-diastereoisomers, should be added for clarity of the readers.
This manuscript is a resubmission of an earlier submission. The following is a list of the peer review reports and author responses from that submission.
Round 1
Reviewer 1 Report
The manuscript by Anastasiya L. Khandazhinskaya et al. describes the synthesis and antiviral properties of several 5'-norcarbocyclic derivatives of substituted 5-arylamino- and 5-aryloxyuracils. Unfortunately, neither novelty of compounds nor their properties make the paper suitable for publishing in Int. J. Mol. Sci.
All compounds were prepared by known methods, according to cited literature, with some improvements. The synthetic part does not contain any substantial achievements.
Several of the prepared compounds showed some antiviral activity; however, significantly lower than those of already known antiviral agents. Thus, I do not agree that the presented compounds ‘are promising for a more detailed study’. Moreover, there is no rationale for the choice of substituents in position 5. It seems that they are random, according to the availability of phenols and amines.
Thus, I recommend rejecting the paper. It should be submitted to a more specialized journal, like Nucleosides, Nucleotides and Nucleic Acids.
Author Response
The authors are grateful to the reviewer for the time spent and the expressed point of view, but consider it necessary to present their arguments in favor of publishing this article in special issue “State-of-the-Art Biochemistry in Russia”of Int. J. Mol. Sci.
Speaking about “novelty of compounds” we should stress that all the compounds described in the paper are original and were synthesized and characterized first time. Previously, we described methods for the preparation of 5-arylamino derivatives of uracil [Bioorg. Med. Chem. 2010; 18 (23), 8310-8314], however, in this work, not only new derivatives of this type were obtained, but also another type of compounds - 5-aryloxyuracils. The possibility of synthesizing this group of compounds by the methods developed by us earlier was not obvious.
The authors agree with the reviewer that the detected activity is moderate, however we would like to note that this work is the first example of the activity of 5’-norcarbocyclic analogs of 2’,3’-dideoxy-2’,3’-didehydrouridine against RNA viruses. It can be assumed that the synthesized analogs act according to a mechanism different from the usual action of nucleoside inhibitors of polymerases, since 5’-norcarbocyclic analogs cannot be phosphorylated by cellular kinases and converted into the corresponding terminator substrates of viral polymerases. Previously, we [Bioorg. Med. Chem. 2010; 18 (23), 8310-8314; Acta Naturae, 2015, 7 â„– 3 (26), 113-115; Chem Biol Drug Des. 2015 Dec;86(6):1387-96] and other groups [Helvetica, 1983, Vol. 66 (2), 534-541] have shown that derivatives of 5-arylaminouracil are able to suppress various pathogens, in particular M. Tuberculosis, HIV, EBV. Our previous studies of the 5-arylamino moiety substituents have shown that the 4-BuO fragment was good choice for antimycobacterial activity; the presence of two methyl residues at positions 3 and 5 gave anti-HIV activity, and 3-PhO-substituent led to activity against Epstein-Barr virus. Phenoxypyrimidinones were described as bronchodilators and antiulcer agents [J. Med. Chem. 1980, 23 (9), 1026-1031]. In connection with the above, the purpose of this work was to evaluate the contribution of the different arylamino- and aryloxy- substituent in the 5 position of uracil to the SAR for RNA viruses.

Reviewer 2 Report
In this manuscript the authors have synthesised a series of new derivatives of 5-substituted uracils, specifically 5'-norcarbocyclic derivatives of substituted 5-arylamino- and 5-aryloxyuracils, and performed preliminary testing of their biological activity against some RNA viruses. The authors have performed a thorough characterisation of the synthesised compounds. I suggest that the manuscript would be suitable for publication in International Journal of Molecular Sciences if the text is subjected to some minor changes, as follows: Line 18. Insert “the” before “herpes” Line 23. Define “EBV” Lines 41-42. Change “Nucleic acid” to “NA” Line 45. Insert “the” before “herpes” Line 50. Change “And only” to “Only” Line 61. Define “EBV” Lines 62-63. Define the terms “HCV, VSV, HSV-1, HSV-2, mCMV, and TBEV” Line 76. Italicise “M. tuberculosis”, “M. avium”, and “M. bovis” Line 81. Italicise “M. tuberculosis” Line 84. Define “MDR” and italicise “M. tuberculosis” Line 230. Change “was” to “were” Line 263. Change “8,48” to “8.48” Line 318. Change “Ethyl ester” to “The ethyl ester” and make bold “6a-d” Lines 318-319. Provide the quantities of “ethyl formate (111 mmol) and NaH (113 mmol,” Line 322. Provide the quantities of “thiourea (67.00 mmol) and NaH (66.67 mmol,” Line 323. Change “latter” to “later” Line 351. Change “Corresponding” to “The corresponding” Line 361. Make bold “3a” Line 364. Insert “a” before “yellow” Line 372. Insert “a” before “yellow” Line 395. Insert “a” before “yellowish” Line 546. Make bold “3i” Line 660. Change “reducing” to “to reduce” Line 669. Change “reducing” to “to reduce” Line 678. Insert “the” before “Russian” Line 679. Insert “the” before “Russian”Author Response
The authors are grateful to the editor and reviewers for their constructive criticism and helpful advice. We appreciate the reviewer’s comments and have fixed all the typos.

Round 2
Reviewer 1 Report
The revised version of the manuscript does not contain any substantial improvements. The explanation by the authors does not change my opinion that the studies described in the manuscript would not be interesting to the readers of IJMS. The chemical part is routine, while the biological results are not especially promising. Thus, I confirm my recommendation to reject the manuscript to keep the high level of the journal.